# Single-cell analysis of a high-grade serous ovarian cancer cell line reveals transcriptomic changes and cell subpopulations sensitive to epigenetic combination treatment

**Shruthi Sriramkumar**[1], **Tara X. Metcalfe**[1], **Tim Lai**[2,3], **Xingyue Zong**[1], **Fang Fang**[4], **Heather M. O'Hagan**[1,4,5]*, **Kenneth P. Nephew**[1,5,6]*

**1** Cell, Molecular and Cancer Biology Graduate Program and Medical Sciences Program, Indiana University School of Medicine, Bloomington, Indiana, United States of America, **2** Luddy School of Informatics, Computing, and Engineering, Indiana University, Bloomington, Indiana, United States of America, **3** Department of Mathematics, Indiana University, Bloomington, Indiana, United States of America, **4** Department of Medical and Molecular Genetics, Indiana University School of Medicine, Indianapolis, Indiana, United States of America, **5** Indiana University Melvin and Bren Simon Comprehensive Cancer Center, Indianapolis, Indiana, United States of America, **6** Department of Anatomy, Cell Biology and Physiology; Department of Obstetrics and Gynecology, Indiana University School of Medicine, Indianapolis, Indiana, United States of America

* knephew@indiana.edu (KPN); hmohagan@indiana.edu (HMO)

**Data Availability Statement:** Single-cell RNA-seq data generated in this study are available through

## Abstract

Ovarian cancer (OC) is a lethal gynecological malignancy with a five-year survival rate of only 46%. Development of resistance to platinum-based chemotherapy is a common cause of high mortality rates among OC patients. Tumor and transcriptomic heterogeneity are drivers of platinum resistance in OC. Platinum-based chemotherapy enriches for ovarian cancer stem cells (OCSCs) that are chemoresistant and contribute to disease recurrence and relapse. Studies examining the effect of different treatments on subpopulations of HGSOC cell lines are limited. Having previously demonstrated that combined treatment with an enhancer of zeste homolog 2 inhibitor (EZH2i) and a RAC1 GTPase inhibitor (RAC1i) inhibited survival of OCSCs, we investigated EZH2i and RAC1i combination effects on HGSOC heterogeneity using single cell RNA sequencing. We demonstrated that RAC1i reduced expression of stemness and early secretory marker genes, increased expression of an intermediate secretory marker gene and induced inflammatory gene expression. Importantly, RAC1i alone and in combination with EZH2i significantly reduced oxidative phosphorylation and upregulated Sirtuin signaling pathways. Altogether, we demonstrated that combining a RAC1i with an EZH2i promoted differentiation of subpopulations of HGSOC cells, supporting the future development of epigenetic drug combinations as therapeutic approaches in OC.

## Introduction

Ovarian cancer (OC) is the fifth leading cause of death among U.S. women [1]. High grade serous OC (HGSOC) accounts for 70% of epithelial ovarian cancer (EOC) [1]. The standard of

NCBI's Gene Expression Omnibus (GEO) through GEO series accession number GSE207993.

**Funding:** This research was funded in part by the Ovarian Cancer Research Alliance (grant number 458788 to HMOH and KPN), the Ovarian Cancer Alliance of Greater Cincinnati (to KPN) and through the IU Simon Comprehensive Cancer Center P30 Support Grant (P30CA082709-20). SS was supported by the Doane and Eunice Dahl Wright Fellowship generously provided by Ms. Imogen Dahl. The funders had no role in study design, data collection and analysis, decision to publish, or preparation of the manuscript.

**Competing interests:** The authors have declared that no competing interests exist.

care for OC patients is debulking surgery followed by platinum-taxane based chemotherapy [2]. Although most patients are initially responsive to chemotherapy, many patients develop tumor recurrence and relapse that rapidly evolves into chemoresistant disease, which is universally fatal [3]. EOC, like most solid tumors, is composed of a diverse array of cell types and it is well established that tumor heterogeneity plays a key role in the progression of EOC [4]. Tumor heterogeneity including transcriptomic heterogeneity also contributes to therapy resistance [5]. Therefore, a better understanding of the tumor subpopulations in general, as well as the effects of treatments on different cell populations in the tumor, is imperative.

In OC, data from preclinical models as well as patient samples have strongly established that platinum-based chemotherapy enriches for drug resistant aldehyde dehydrogenase positive (ALDH+) ovarian cancer stem cells (OCSCs) [6] that contribute to tumor relapse and disease recurrence [7]. We and others have identified several targets that can be exploited to block the platinum driven enrichment of ALDH+ OCSCs [6, 8–13]. In a recent study, we demonstrated that combination treatment of enhancer of zeste homolog 2 inhibitor (EZH2i) and an inhibitor targeting small GTPase RAC1 inhibited the survival of ADLH+ OCSCs *in vitro* [9]. In addition, we demonstrated that this combination treatment inhibited tumor growth and increased the sensitivity of HGSOC cells to platinum agents *in vivo* [9]. Although transcriptomic analysis of whole-cell populations showed genes and pathways altered by coadministration of EZH2i and RAC1i [9], it is likely that the treatments also altered both the number of cells and gene expression in subpopulations of cells within the whole-cell population.

Traditional transcriptomic analysis in biomedical research using bulk RNA-seq is state-of-the-art for studying gene expression and trends of whole cell populations. However, to detect biologically meaningful changes in gene expression in subpopulations of cells that make up a small percentage of the total tumor population, single cell RNA-sequencing (scRNA-seq) is currently used. ScRNA-seq is capable of capturing the transcriptomics of each individual cell and provides meaningful insight when used in primary samples and cancer cell lines [14–17]. scRNA-seq has the potential to identify key differences in cellular phenotypes and uncover molecular mechanisms that regulate tumor heterogeneity [18, 19]. For example, using scRNA-seq of primary, relapsed and metastatic OC, CYR61+ cells were identified that can be used as a biomarker of EOC recurrence and could be a therapeutic target [20]. However, to date, scRNA-seq analysis of cell lines representing HGSOC is lacking, and knowledge on baseline transcriptomes and treatment-induced OC cell line heterogeneity at the single cell level is limited.

In the current study, we utilized scRNA-seq to further investigate the effect of combining inhibitors of EZH2 and RAC1 (EZH2i and RACi) on OC cell heterogeneity. With the high resolution of the scRNA-seq transcriptomic analysis, we identified key subpopulations related to OCSCs. We showed that RAC1i treatment either alone or combined with EZH2i significantly increased the number of cells in subpopulations associated with cell differentiation and development and changed gene expression of stemness and secretory markers. Furthermore, the scRNA-seq analysis identified a subpopulation of cells marked by expression of inflammatory genes, which were more uniformly expressed following treatment with RAC1i. This study is the first report of single-cell analysis of a HGSOC cell line and the impact of epigenetic and small GTPase inhibitor combination therapy on HGSOC subpopulations.

## Materials and methods

### Cell culture

OVCAR3, a representative HGSOC cell line, was maintained at 37˚C and 5% $CO_2$ as described previously [6, 9]. This cell line was authenticated by ATCC in 2018. OVCAR3 cells were

cultured in DMEM 1X (Gibco, #11995065) containing 10% FBS (R&D Systems, #S11150) without antibiotics. OVCAR3 cells used in the study were passaged for less than 15 passages. 50 mM stock solution of RAC inhibitor (NSC23766, Sigma #553502) was made in DMSO. 5 mM stock solution of EZH2 inhibitor (GSK126, Biovision #2282) was made in DMSO. For all the experiments using these inhibitors, an equivalent amount of DMSO or RAC1i or EZH2i or combination of both was added to cells and incubated for 48 hrs at 37˚C and 5% $CO_2$.

## Antibodies

Anti-OVGP1 (Abcam, #ab74544), Anti-Ki-67 (Cell Signaling Technology (CST), #9449S), Alexa- Fluor 488 (CST, #4412), Alex-Fluor 594 (CST, #8890).

## Single cell RNA-seq

Approximately 10,000 cells per sample were targeted for input to the 10X Genomics Chromium system using the Chromium Next GEM Single Cell 3' Kit v3.1 at the Indiana University School of Medicine (IUSM) Center for Medical Genomics satellite core in Bloomington. The libraries were sequenced at the IUSM Center for Medical Genomics using a NovaSeq 6000 with a NovaSeq S2 reagent kit v1.0 (100 cycles) with approximately 450 million read pairs per sample. To generate the count matrices, 10X Genomics Cell Ranger (v4.0.0) with default settings and genome assembly, GRCh38 (2020-A) were used. The scripts used for the scRNA-seq analysis are available on Github (https://github.com/timlai4/agnes_scRNA). The resulting matrices were inputted into Seurat (v3.1.5) for further processing, including batch correction using canonical correlation-based alignment analysis. The quality of the data was assessed using various metrics, including the ratio of mitochondrial transcript content to the number of detected genes for each cell. Cells with low gene counts (less than or equal to 1000 genes), high mitochondrial content (greater than 20% mtRNA), or low complexity score (less than or equal to 0.8) were removed, resulting in approximately 7500–10000 remaining cells per sample. Using the built-in Seurat function *CellCycleScoring*, the cell-cycle stages were determined through analysis of pre-annotated genes. UMI counts of the remaining cells were normalized using SCTransform [21] regressing out cell cycle and mitochondrial effects. Dimension reduction via Principal Component Analysis (PCA) [22] was then performed on the integrated data followed by a Leiden-based clustering method [23] (Wilcoxon rank-sum test; FDR < 0.05 and Log Fold Change > 0.4). To visualize the clusters, we used UMAP [22] dimension reduction to display the points in 2D.

To compare the proportional differences in cell populations between two conditions, we used *scProportionTest* an R library [14] (https://github.com/rpolicastro/scProportionTest/releases/tag/v1.0.0).

## JC-1 staining

$1 \times 10^6$ OVCAR3 cells were cultured in 100 mm plates for 48 hours. After 48 hours, cells were treated with respective doses of DMSO, 50 μM RAC1i, 5 μM EZH2i or combination of RAC1i and EZH2i for 48 hours. Following the incubation, JC-1 staining was performed as the manufacturer's instructions. JC-1 (Thermo Fisher, #T3168) stock solutions were made in DMSO at 5 mg/ml concentration. Appropriate amount of JC-1 was added to cells such that the final concentration was 2 μg/ml and the samples were incubated for 30 minutes at 37˚C and 5% $CO_2$. After the incubation, cells were collected and analyzed by flow cytometry as described previously [6].

## Flow cytometry

LSR II Flow Cytometer (BD Biosciences) was used for the analysis of JC-1stained samples. J-monomers and J-aggregates was measured using 488nm excitation and the signal was detected using AF488 (green) and PE-A (red), respectively.

## Immunofluorescence

$2 \times 10^5$ OVCAR3 cells were cultured in 6-well plates on coverslips and incubated for 48 hours. After 48 hours, cells were treated with EZH2i, RAC1i, combination of EZH2i and RAC1i or an equivalent amount of DMSO for 48 hours and then immunofluorescence was performed. Cells were fixed with 4% paraformaldehyde in PBS. Following fixation, cells were permeabilized with 0.5% Triton-X in PBS and blocked with 1% BSA in PBST (PBS + 0.1% Tween 20). After blocking cells were incubated with anti-OVGP1 (Abcam, #ab74544) and Ki-67 (CST, #9449S) for 1 hour at RT. This was followed by incubation with Alex fluor conjugated secondary antibodies. Prolong Gold Antifade with DAPI (CST, #8961) was used for mounting coverslips.

## Imaging

All the images were acquired using the Leica SP8 scanning confocal system with the DMi8-inverted microscope and Leica LASX software (Leica Microsystems). Images were taken using 63X, 1.4NA oil immersion objective at room temperature. Images were processed using Image J (National Institutes of Health, Bethesda, MD).

# Results

## Single cell RNA-sequencing identifies subpopulations of cells in a HGSOC cell line

To better understand the effects of the RAC1i and EZH2i on subpopulations of HGSOC cells, we performed scRNA-seq using the HGSOC cell line OVCAR3 (Fig 1). Following batch correction and dimension reduction, eleven cell clusters were identified and visualized using UMAP (Fig 1A). For each cluster, the most significant markers across all the samples were generated (see S1 Table for top genes enriched in each cluster). Metascape analysis was performed on marker genes positively associated with the different clusters to identify pathway enrichment [24] (Fig 1B–1E, S2 Table). Metascape analysis revealed that cluster 2 was enriched in genes involved in development and differentiation and cluster 5 was enriched in immune response markers, while cluster 4 and 8 showed enrichment of cell cycle and DNA repair genes (Fig 1B–1D). Interestingly, cluster 6 and 7 showed enrichment of genes involved in mitochondrial oxidative phosphorylation (OXPHOS) (Fig 1E). Clusters 1, 3, 9 and 10 were mostly defined by negative enrichment of certain genes within the given cluster relative to the other clusters and therefore could not be assessed by Metascape analysis. These subpopulations were subsequently assigned as miscellaneous (misc.) clusters.

## RAC1 inhibition induces changes in the proportion of cells in different clusters

UMAP visualization revealed that all clusters were present in each treatment group but differed in relative proportion (Fig 2A and 2B, S3 Table). Treatment with 50 µM RAC1i alone induced significant changes in the proportion of cells in clusters 2:Cell differentiation and development (CDD) and 10:Misc 4 compared to DMSO treated cells (Fig 2C). RAC1i and

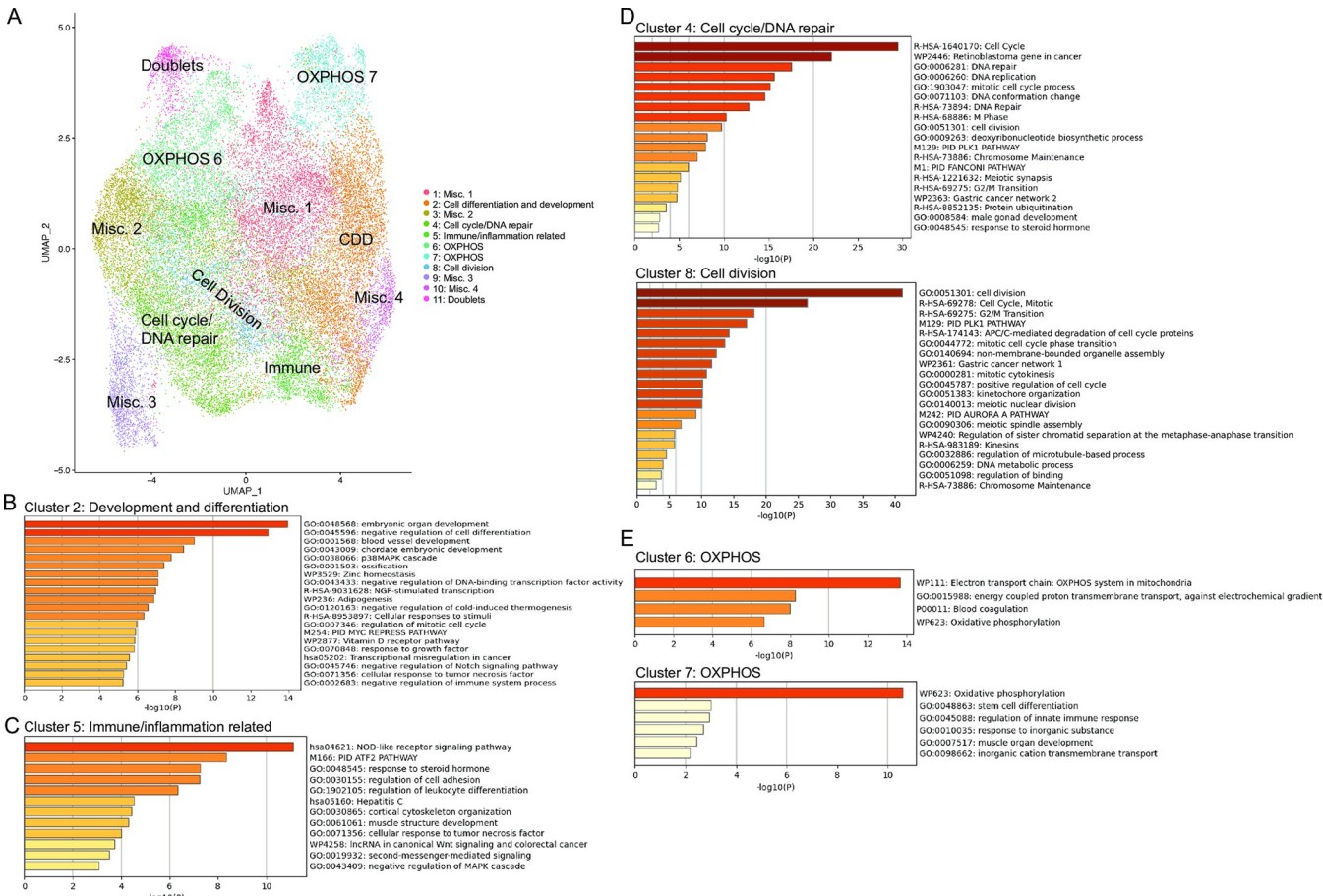

**Fig 1. Single cell analysis identifies cell subpopulations in a HGSOC cell line.** (A) Uniform Manifold Approximation and Projection for Dimension reduction (UMAP) dot plot of cells from all samples colored by cluster. (B-E) Enriched pathways identified by Metascape analysis of positively associated marker genes in clusters 2 (B), 5 (C), 4 and 8 (D), and 6 and 7 (E).

EZH2i (5 μM) combination induced similar changes in cluster 2:CDD and 10:Misc 4 relative to DMSO and decreased the proportion of cells in cluster 9:Misc. 3. However, no significant changes to the proportion of cells in the various clusters were observed after EZH2i treatment alone (Fig 2B). In the CDD cluster, treatment with RAC1i alone or in combination with EZH2i increased the cell proportions from 7% to approximately 25% of the entire cell population while EZH2i alone had no effect on this cluster (Fig 2B), suggesting that RACi treatment altered differentiation.

## RAC1 inhibition induces uniform expression of inflammatory marker genes

Cluster 5 was enriched for expression of genes involved in the immune response. Because inflammation and the immune response contribute to EOC, we examined this cluster more closely and identified the genes that were driving this classification, including chemokines, *CXCL1*, *CXCL2*, *CXCL3*, and *CXCL8*. UMAP blots for these genes indicated that in DMSO and EZH2i only samples, these genes were predominantly only expressed in cluster 5, our identified immune cluster (Fig 3). Following RAC1i treatment alone or in combination with EZH2i, *CXCL1* expression increased in cluster 5 but became more apparent in the other

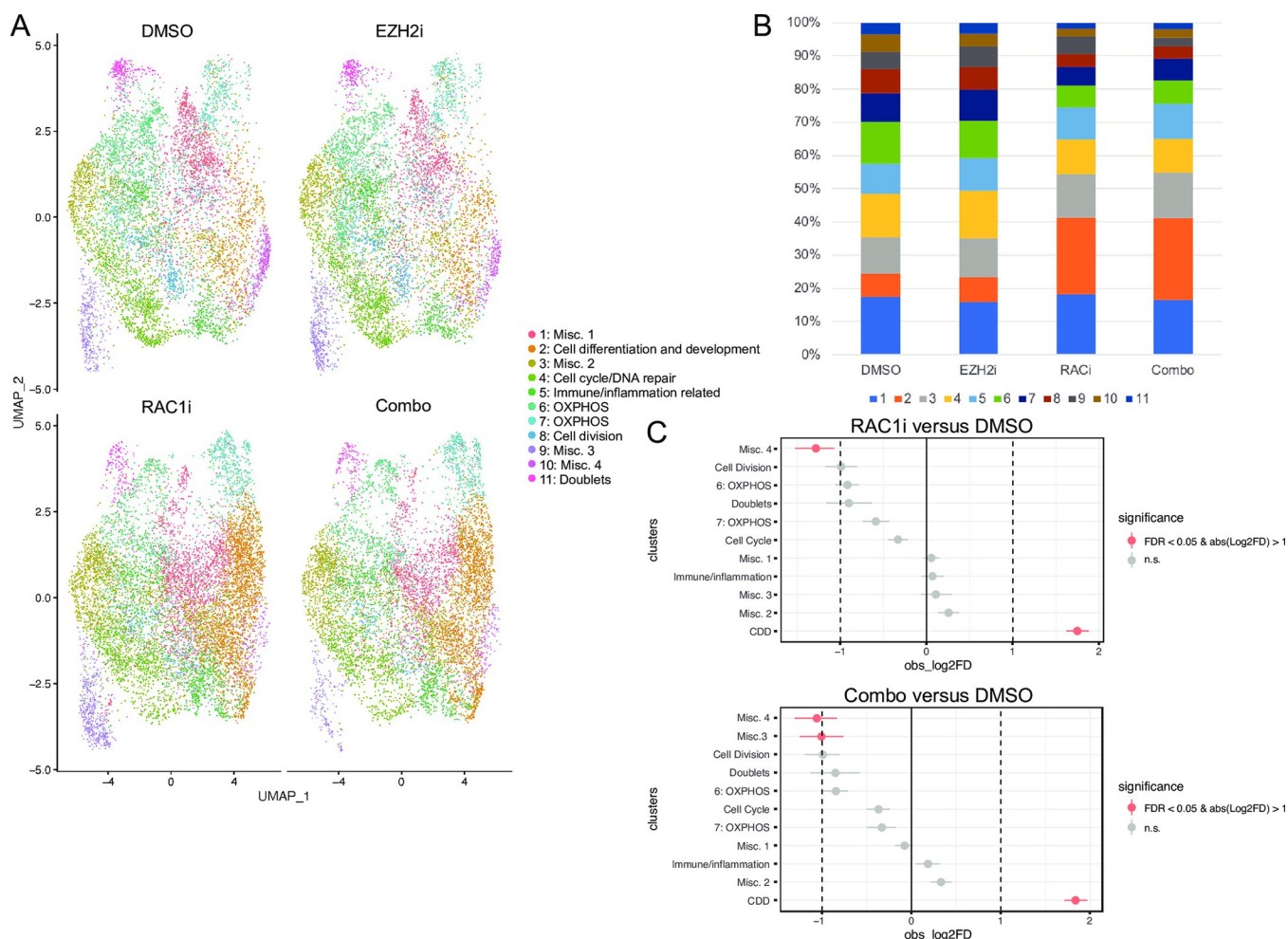

**Fig 2. RAC1i treatment changes the relative proportion of cells in different clusters.** (A) Individual UMAP dot plots of DMSO, EZH2i (GSK126, 5 μM, 48H), RAC1i (NSC23766, 50 μM, 48H), and Combo (RAC1i + EZH2i) scRNA-seq samples colored by cluster. (B) Relative proportion of cells in each cluster for each sample type. (C) Relative differences in cell proportions for each cluster between RAC1i versus DMSO and combo treatment versus DMSO. Red clusters have an FDR < 0.05 and mean | Log2 fold enrichment | > 1 compared with the normal colon (permutation test; n    10,000).

clusters as well (Fig 3A). *CXCL2*, *CXCL3*, and *CXCL8* expression also increased in cluster 5 following RAC1i treatment, with some additional positive cells in the other clusters as well (Fig 3B–3D). The increase in CXCL1, but not CXCL2 or CXCL3, expression was detectable in bulk OVCAR3 cells (S1A Fig). This data suggests that even though RAC1i treatment did not alter the proportion of cells in cluster 5, it did increase expression of marker genes associated with cluster 5 both in that cluster and in other cells in the population.

## RAC1i treatment decreases oxidative phosphorylation

As the proportion of cells in cluster 2:CDD after RAC1i and combination treatment markedly increased, it was of interest to identify pathways in this cluster that changed with treatment. To this effect, we compared cluster 2 gene expression in EZH2i, RAC1i and combination treatment to the control. IPA analysis revealed that RAC1i and combination treatment resulted in significant downregulation of oxidative phosphorylation (OXPHOS), cholesterol biosynthesis and estrogen receptor signaling and upregulation of the Sirtuin signaling pathways in cluster 2

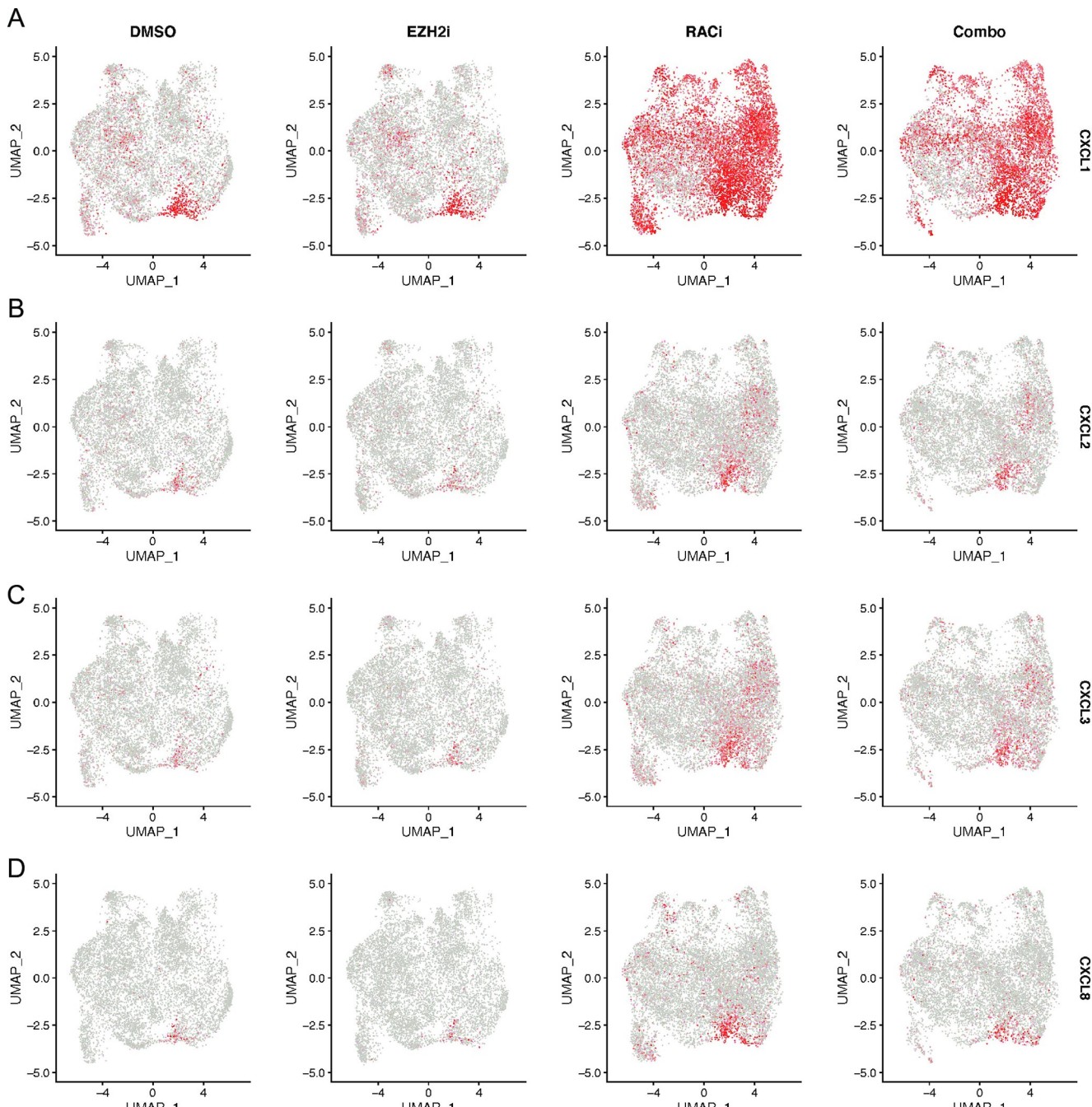

**Fig 3. RAC1i treatment results in broader expression of inflammatory genes.** (A-D) UMAP dot plots of normalized expression values of CXCL1 (A), CXCL2 (B), CXCL3 (C) and CXCL8 (D) in the indicated sample types.

when compared to cluster 2 DMSO treated samples (Fig 4A, S4 Table). In contrast, cholesterol biosynthesis was upregulated in cluster 2 following EZH2i treatment alone compared to DMSO (Fig 4A, S4 Table). EIF2 signaling was increased and decreased in EZH2i and RAC1i single treatment samples, respectively, and not altered in the combination treatment cluster 2 cells. To validate the decrease in OXPHOS following RAC1i treatment, we used JC-1, a lipophilic cationic dye, which is commonly used to measure mitochondrial membrane potential

($\Delta\Psi_M$), a surrogate for OXPHOS [25]. In cells with normal $\Delta\Psi_M$ and healthy mitochondria, JC-1 will spontaneously aggregate and form red fluorescent J-aggregates; however, in cells with defective $\Delta\Psi_M$, JC-1 does not form aggregates and retains its green fluorescence [26]. Treatment of HGSOC cell line OVCAR3 with RAC1i for 48 hours followed by JC-1 increased green fluorescence intensity (Fig 4B), consistent with our IPA analysis. In the identified Sirtuin signaling pathway enriched following RAC1i treatment, *SIRT7* was the most highly upregulated gene. Therefore, we examined *SIRT7* expression across the different clusters and treatment groups. Treatment with RAC1i alone or in combination with EZH2i induced *SIRT7* expression when compared to treatment with EZH2i alone or DMSO in all cell clusters (Fig 4C). UMAP plots and RT-qPCR of bulk OVCAR3 cells also confirm the induction of *SIRT7* in RAC1i only and combination treatment compared to EZH2i treatment alone or DMSO (Fig 4C, S1B Fig).

## RAC1 inhibition induces differentiation of OC cells

Previously, we had demonstrated that RAC1i and EZH2i combination treatment reduced the ALDH+ population of OCSCs [9]. Therefore, we were interested in determining if combination treatment had any cell cluster-specific effects on the expression of key genes associated with OCSCs. *ALDH1A1* was predominantly expressed in clusters 3–9 and 11 in DMSO treated cells (Fig 5A). *CD24* and *SOX2* were more uniformly expressed across all clusters in DMSO treated cells. In all clusters in which these genes were expressed, treatment with RAC1i alone or in combination with EZH2i reduced the expression of *ALDH1A1*, *SOX2*, *CD24* (Fig 5A). Changes in *CD24* expression was more variable between clusters than *ALDH1A1* and *SOX2*. Additional genes commonly associated with OCSCs were either expressed at low levels (CD44, PROM1/CD133, ALDH1A2, ALDH1A3) and therefore expression changes with treatment were unable to be determined (S2 Fig). The expression of other expressed ALDH genes (ALDH2, ALDH6A1, ALDH7A1, ALDH9A1) also decreased across most clusters with RAC1i treatment alone or in combination with EZH2i (S2 Fig). Because our initial analysis had identified changes in cluster 2 with RAC1i treatment, which is enriched for genes associated with cell development and differentiation (Fig 2C), we also examined expression of *PAX8*, an early secretory marker commonly expressed in EOC [27, 28]. *PAX8* expression was uniform in control DMSO and EZH2i samples. However, in samples treated with RAC1i alone or in combination, *PAX8* expression decreased across all clusters (Fig 5B, top panels). Next, we examined the expression of *OVGP1*, a marker for intermediate secretory cells reported to be expressed in OC [29, 30]. Expression of *OVGP1* in DMSO and EZH2i alone samples was low in all clusters. However, treatment with RAC1i or combination of RAC1i and EZH2i induced expression of *OVGP1* across all clusters (Fig 5B, bottom panels). To validate this finding, we treated OVCAR3 cells with EZH2i or RAC1i alone or in combination and performed immunofluorescence for OVGP1 and proliferation marker Ki-67. Consistent with our single cell data, treatment with RAC1i alone or in combination with EZH2i induced the expression of OVGP1 (Fig 5C). OVGP1 stained cells co-stained with Ki-67 indicated that OVGP1 expression was induced in proliferating cells (Fig 5C). Altogether, this data suggests that stemness markers are expressed by most cells in the HGSOC cell line OVCAR3 and that RAC1i alone or in combination caused the cells to become more differentiated, based on reduced expression of stemness markers and increased expression of OVGP1, an intermediate secretory cell marker.

## Discussion

Single cell transcriptomics provides a powerful means of investigating changes in gene expression at the level of individual cells. While traditional bulk RNA-sequencing gives an overall

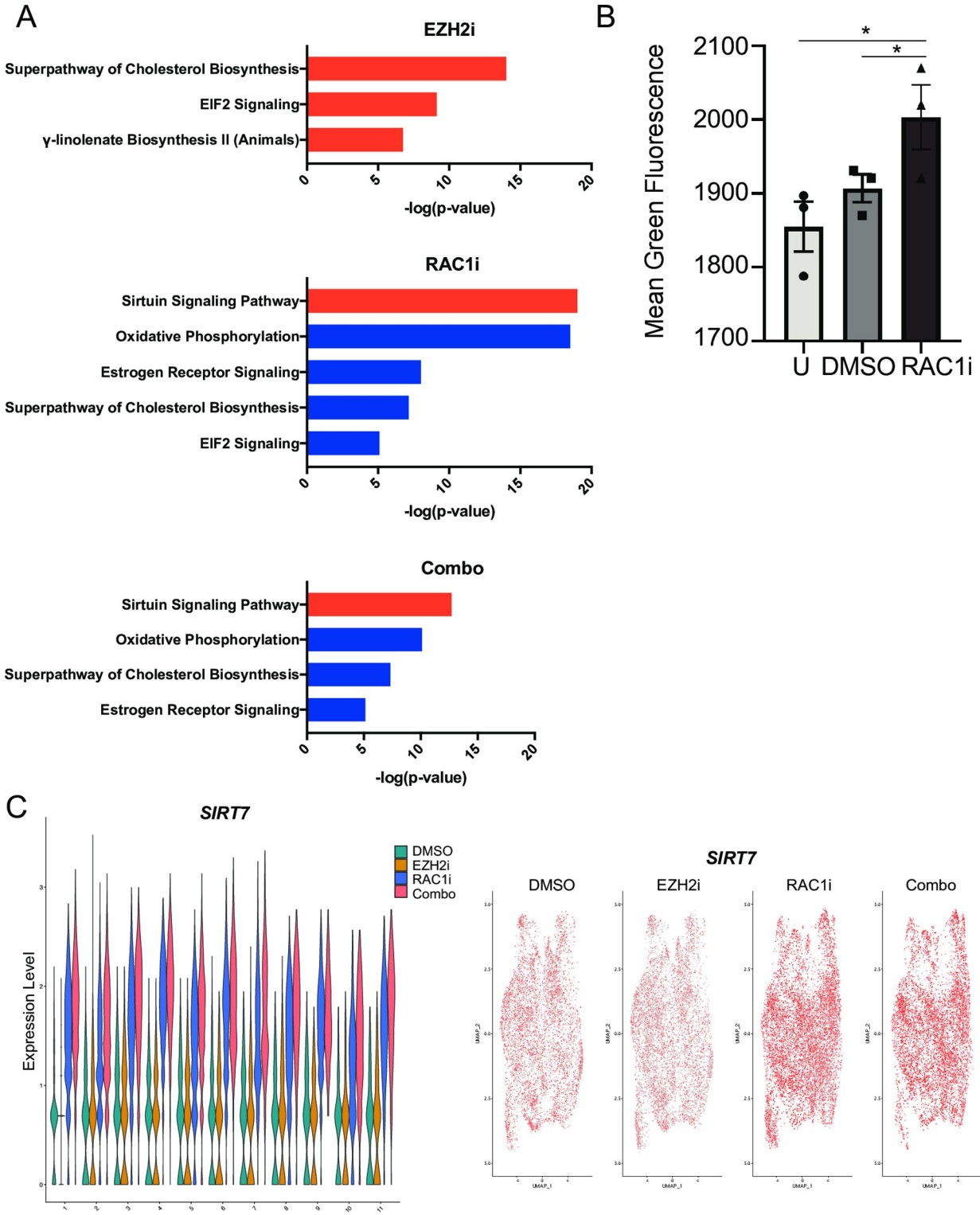

**Fig 4. RAC1 inhibition results in a decrease in oxidative phosphorylation.** (A) Enriched pathways identified by IPA of gene expression alterations in cluster 2 of the indicated treatment compared to DMSO. Orange or blue bars indicate a positive or negative activation z-score, respectively, as calculated by IPA based on expected directionality of expression changes. (B) Graph of mean green fluorescence intensity of cells untreated (U) or treated with DMSO and RAC1i (50 μM, 48H) followed by JC-1 staining for 30 minutes and analysis by flow cytometry. Graph depicts mean -/+ SEM. N = 3. *$P<0.05$. (C) Violin and UMAP dot plots of *SIRT7* expression levels in the different clusters and sample types.

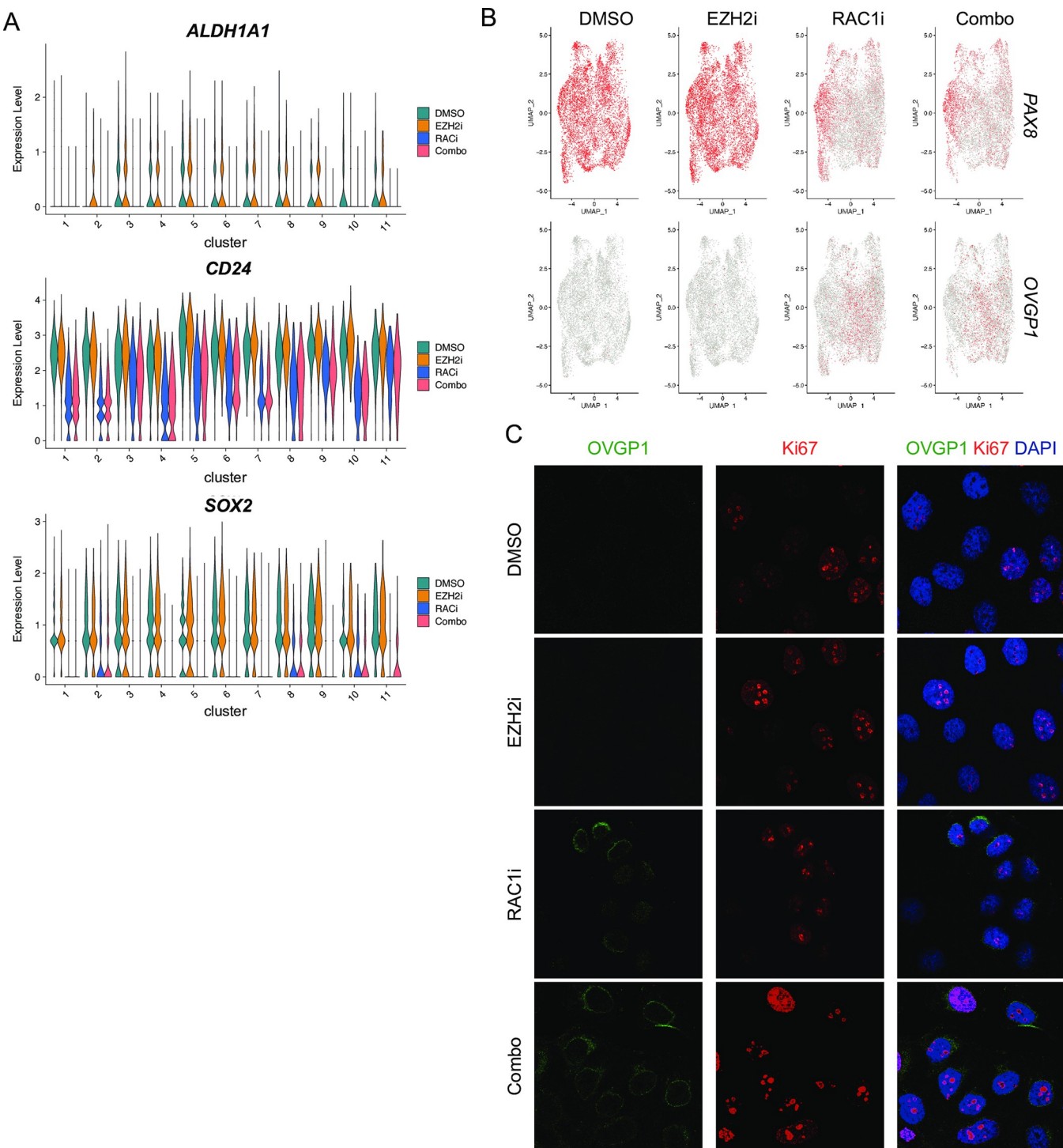

**Fig 5. RAC1 inhibition reduces stem cell marker expression in most clusters.** (A) Violin plots of expression levels of indicated genes in the different clusters and sample types. (B) UMAP dot plots of normalized gene expression values in the indicated sample types. (C) Representative immunofluorescence images of OVGP1 (green) and Ki67 (red) staining after DMSO, EZH2i (5 μM, 48H), RAC1i (50 μM, 48H), or Combo (RAC1i + EZH2i) treatment. N = 3.

picture of transcriptomics throughout the entire sample, effects on individual subpopulations can be difficult to infer. In this regard, based on our previous study using bulk RNA-seq showing epigenetic treatment effects on stemness-associated genes and the WNT signaling pathway

in HGSOC [9], it was of interest to investigate the possible treatment effects at a single cell resolution. To this end, we used scRNA-seq to profile OVCAR3 cells before and after treatment with EZH2i or RAC1i alone or in combination. We identified cell clusters and treatment effects on individual clusters.

Previous studies have identified several classes of marker genes important for assessing OCSCs [31–33]. In this current study, expression levels of well-known OCSC markers, *ALDH1A1*, *CD24* and *SOX2* were used to assess the expression of stemness genes in the clusters. Recently, a study on the fallopian tube epithelium (FTE) using scRNA-seq [34] identified several marker genes of different cell types in the FTE, including secretory cells. *PAX8* and *OVGP1* were identified as marker genes of early and intermediate secretory cells, respectively. Because the FTE is a major site of origin of HGSOC, we also compared expression levels of the secretory marker genes *PAX8* and *OVGP1* across our samples. Uniform expression of *PAX8* and no *OVGP1* expression was seen in all clusters in the control and EZH2i-treated samples, indicative of early secretory cell populations and stemness. Expression levels of *ALDH1A1* tended to be more localized yet still prevalent throughout multiple clusters, and RAC1i treatment significantly reduced expression in all clusters. In contrast, expression of *OVGP1* was only detectable after treatment with RAC1i. The reciprocal changes in *PAX8* and *OVGP1* expression suggest that RAC1i treatment is altering secretory cell fates. As a result, we attempted to trace potential cell fates using the other markers in the study [34] but did not observe this phenomenon with any of the other epithelial markers. Nevertheless, these changes can be explored with trajectory analysis in the future.

In the cell differentiation and development cluster 2, we demonstrate that EZH2i and RAC1i combination treatment inhibited the OXPHOS pathway while activating the Sirtuin signaling pathway. The Sirtuin pathway is related to epithelial-to-mesenchymal transition (EMT) suppression and has been studied in lung and ovarian cancers [35, 36]. In particular, the SIRT1 gene represses EMT and antagonizes migration in vitro and metastases in vivo. On the other hand, OXPHOS is important to cell proliferation [37–39]. Studies have shown that CSCs derive energy from OXPHOS and in the presence of active OXPHOS, CSCs develop antioxidant mechanisms to manage ROS [40–44]. In this context, OC cells would no longer respond to ROS-inducing agents, a key shared feature of many current treatment options. As development of chemoresistance is a major obstacle in the treatment of OC patients, determining if the alteration in OXPHOS caused by treatment with RAC1i and EZH2i combination sensitizes HGSOC cells to platinum-based agents is an important question that warrants further investigation.

In a previous study [9], we showed that treatment targeting EZH2 and RAC1 in combination with platinum chemotherapy inhibits OCSCs. Unexpectedly, in our current study, EZH2 mediated effects were not apparent. The precise reason for this is unclear presently. It is possible that the significant effects seen in the bulk transcriptome analysis was an aggregate of incremental changes in multiple clusters, which individually would not be statistically significant. Due to the current limitations of single-cell protocols, there may not have been enough cells to investigate a few of the target pathways, especially the WNT signaling effects. Nevertheless, we detected significant changes after treatment with the RAC1 inhibitor alone or in combination with EZH2i.

## Conclusions

Although OC cell lines are widely used to test and validate treatments in vitro and in vivo, to date no single cell analysis studies of HGSOC cell lines have been reported in the literature. As such, the current study represents the first exploratory single-cell analysis of a HGSOC cell line

transcriptome and distinguishable subpopulations sensitive to epigenetic combination treatment. Although most of the clusters were distinguished through their biological functions, gene signatures were identified that we believe will drive future research on targeted therapy, including the OXPHOS and Sirtuin pathways.

## Supporting information

**S1 Fig. RAC1 inhibition induces altered expression of some genes in bulk OVCAR3 cells.** A and B) OVCAR3 cells were treated with DMSO, 5 μM. EZH2i, 50 μM RAC1i or combination for 48 hours. cDNA was made from RNA collected from bulk populations of cells and RT-qPCR was performed. Expression of the indicated gene was normalized to a housekeeping gene and then to the untreated control. N = 3. Graphs depict mean +/- SEM. $^*P<0.05$, $^{**}P<0.01$, $^{***}P<0.001$.
(DOCX)

**S2 Fig. RAC1 inhibition reduces expression of several ALDH isoforms in most clusters.** Violin plots of expression levels of indicated genes in the different clusters and sample types.
(DOCX)

**S1 Table. Top genes enriched in each cluster.**
(DOCX)

**S2 Table. Metascape analysis results using genes enriched in each cluster.**
(XLSX)

**S3 Table. Number of cells in each cluster for each treatment group.**
(DOCX)

**S4 Table. IPA analysis of differentially expressed genes in cluster 2 in the indicated comparisons.**
(XLSX)

## Acknowledgments

We thank the Indiana University Flow Cytometry Core Facility and Center for Genomics and Bioinformatics for their assistance.

## Author Contributions

**Conceptualization:** Xingyue Zong, Heather M. O'Hagan, Kenneth P. Nephew.

**Data curation:** Shruthi Sriramkumar, Tara X. Metcalfe, Tim Lai, Fang Fang, Heather M. O'Hagan.

**Formal analysis:** Shruthi Sriramkumar, Tara X. Metcalfe, Tim Lai, Xingyue Zong, Heather M. O'Hagan, Kenneth P. Nephew.

**Funding acquisition:** Heather M. O'Hagan, Kenneth P. Nephew.

**Investigation:** Shruthi Sriramkumar, Tara X. Metcalfe, Tim Lai, Xingyue Zong, Fang Fang, Heather M. O'Hagan, Kenneth P. Nephew.

**Methodology:** Xingyue Zong, Fang Fang.

**Project administration:** Heather M. O'Hagan, Kenneth P. Nephew.

**Resources:** Heather M. O'Hagan, Kenneth P. Nephew.

**Supervision:** Heather M. O'Hagan, Kenneth P. Nephew.

**Validation:** Shruthi Sriramkumar, Heather M. O'Hagan.

**Visualization:** Shruthi Sriramkumar, Tim Lai, Xingyue Zong, Heather M. O'Hagan.

**Writing – original draft:** Shruthi Sriramkumar, Tara X. Metcalfe, Tim Lai, Xingyue Zong, Heather M. O'Hagan, Kenneth P. Nephew.

**Writing – review & editing:** Shruthi Sriramkumar, Tara X. Metcalfe, Tim Lai, Xingyue Zong, Fang Fang, Heather M. O'Hagan, Kenneth P. Nephew.

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
