## [Decision Letter · Decision Letter 0]

12 May 2022

PONE-D-22-09003Single-cell Analysis of a High-grade Serous Ovarian Cancer Cell Line Reveals Transcriptomic Changes and Cell Subpopulations Sensitive to Epigenetic Combination TreatmentPLOS ONE

Dear Dr. Nephew,

Thank you for submitting your manuscript to PLOS ONE. After careful consideration, we feel that it has merit but does not fully meet PLOS ONE’s publication criteria as it currently stands. Therefore, we invite you to submit a revised version of the manuscript that addresses the points raised during the review process.

We look forward to receiving your revised manuscript.

Kind regards,

Lawrence M. Pfeffer, PhD

Academic Editor

PLOS ONE

Journal Requirements:

3. Thank you for stating the following in the Acknowledgments Section of your manuscript: "We thank the Indiana University Flow Cytometry Core Facility for their assistance. This research was funded in part by the Ovarian Cancer Research Alliance (grant number 458788 to HMOH and KPN), the Ovarian Cancer Alliance of Greater Cincinnati (to KPN) and through the IU Simon Comprehensive Cancer Center P30 Support Grant (P30CA082709-20). SS was supported by the Doane and Eunice Dahl Wright Fellowship generously provided by Ms. Imogen Dahl."

Please remove any funding-related text from the manuscript and let us know how you would like to update your Funding Statement. Currently, your Funding Statement reads as follows: "We thank the Indiana University Flow Cytometry Core Facility for their assistance. This research was funded in part by the Ovarian Cancer Research Alliance (https://ocrahope.org/   grant number 458788 to HMOH and KPN), the Ovarian Cancer Alliance of Greater Cincinnati (http://www.cincyovariancancer.org/  KPN) and through the IU Simon Comprehensive Cancer Center P30 Support Grant (https://www.cancer.iu.edu/  P30CA082709-20 KPL). SS was supported by the Doane and Eunice Dahl Wright Fellowship generously provided by Ms. Imogen Dahl."

4. Thank you for stating in your Funding Statement: "We thank the Indiana University Flow Cytometry Core Facility for their assistance. This research was funded in part by the Ovarian Cancer Research Alliance (https://ocrahope.org/   grant number 458788 to HMOH and KPN), the Ovarian Cancer Alliance of Greater Cincinnati (http://www.cincyovariancancer.org/  KPN) and through the IU Simon Comprehensive Cancer Center P30 Support Grant (https://www.cancer.iu.edu/  P30CA082709-20 KPL). SS was supported by the Doane and Eunice Dahl Wright Fellowship generously provided by Ms. Imogen Dahl."

5. Thank you for stating the following financial disclosure: "We thank the Indiana University Flow Cytometry Core Facility for their assistance. This research was funded in part by the Ovarian Cancer Research Alliance (https://ocrahope.org/   grant number 458788 to HMOH and KPN), the Ovarian Cancer Alliance of Greater Cincinnati (http://www.cincyovariancancer.org/  KPN) and through the IU Simon Comprehensive Cancer Center P30 Support Grant (https://www.cancer.iu.edu/  P30CA082709-20 KPL). SS was supported by the Doane and Eunice Dahl Wright Fellowship generously provided by Ms. Imogen Dahl."

7. We note that you have stated that you will provide repository information for your data at acceptance. Should your manuscript be accepted for publication, we will hold it until you provide the relevant accession numbers or DOIs necessary to access your data. If you wish to make changes to your Data Availability statement, please describe these changes in your cover letter and we will update your Data Availability statement to reflect the information you provide.

8. Please note that in order to use the direct billing option the corresponding author must be affiliated with the chosen institute. Please either amend your manuscript to change the affiliation or corresponding author, or email us at plosone@plos.org with a request to remove this option.

9. PLOS requires an ORCID iD for the corresponding author in Editorial Manager on papers submitted after December 6th, 2016. Please ensure that you have an ORCID iD and that it is validated in Editorial Manager. To do this, go to ‘Update my Information’ (in the upper left-hand corner of the main menu), and click on the Fetch/Validate link next to the ORCID field. This will take you to the ORCID site and allow you to create a new iD or authenticate a pre-existing iD in Editorial Manager. Please see the following video for instructions on linking an ORCID iD to your Editorial Manager account: https://www.youtube.com/watch?v=_xcclfuvtxQ

10. We note that you have included the phrase “data not shown” in your manuscript. Unfortunately, this does not meet our data sharing requirements. PLOS does not permit references to inaccessible data. We require that authors provide all relevant data within the paper, Supporting Information files, or in an acceptable, public repository. Please add a citation to support this phrase or upload the data that corresponds with these findings to a stable repository (such as Figshare or Dryad) and provide and URLs, DOIs, or accession numbers that may be used to access these data. Or, if the data are not a core part of the research being presented in your study, we ask that you remove the phrase that refers to these data.

11. Please upload a new copy of Figures 1, 2, 4, and 5 as the detail is not clear. Please follow the link for more information: https://blogs.plos.org/plos/2019/06/looking-good-tips-for-creating-your-plos-figures-graphics/" https://blogs.plos.org/plos/2019/06/looking-good-tips-for-creating-your-plos-figures-graphics/

Additional Editor Comments (if provided):

The manuscript requires major revisions before it should undergo further review for PLOS. Both reviewers expressed substantial concerns that should be addressed in a revised manuscript.

Reviewers' comments:

Reviewer's Responses to Questions

**Comments to the Author**

1. Is the manuscript technically sound, and do the data support the conclusions?

Reviewer #1: Partly

Reviewer #2: Partly

2. Has the statistical analysis been performed appropriately and rigorously? 

Reviewer #1: I Don't Know

Reviewer #2: I Don't Know

3. Have the authors made all data underlying the findings in their manuscript fully available?

Reviewer #1: Yes

Reviewer #2: No

4. Is the manuscript presented in an intelligible fashion and written in standard English?

Reviewer #1: Yes

Reviewer #2: Yes

5. Review Comments to the Author

Reviewer #1: Overview

Nephew, et al., present data on RNA expression in single cells from the Ovcar3 cell line comparing untreated cells to cells treated with inhibitors of EZH2 and RAC1. This work is a follow up to their previous publication that identified DAB2IP as a tumor suppressor that blocks ovarian cancer stem cells, based on cell line models.

Interesting findings reported include

No major changes in gene expression after treating with the EZH2 inhibitor for 48 hours (Fig 2B). This is quite surprising, as one might expect large scale gene expression changes after blocking this histone methyltransferase, which would then cause differential clustering. (eg, Tiffen, et al., Oncotarget 2015)

RAC1 inhibition has remarkable effects causing upregulation of chemokines, SIRT77 and OVGP1, while downregulating ox/phos genes, Pax8 and stem cell markers ADLH1A1, CD24 and SOX2 in OVCAR3 cell line.

The findings are interesting, although their implications for ovarian cancer are tenuous until they are confirmed in more cell lines and in primary ovarian cancer tissue.

Methods summary

Authors treated the Ovcar3 cell line for 48 hrs with DMSO (control), NSC23766 (RAC1 inhibitor), GSK126 (EZH2 inhibitor), or both inhibitors together and then performed single cell RNAseq (scRNAseq) using the 10X Genomics platform. Data was analyzed using CellRanger and Seurat R packages. Data from ~7 to 10k cells from each treatment were combined and analyzed together. Cells were also treated with JC-1 and analyzed by IFC. The Ovcar3 cell line is a hypotriploid, poorly differentiated papillary adenocarcinoma derived from the ascites of a patient who had been treated with cyclophosphamide, adriamycin, and cisplatin 8 months prior to cell collection in 1982 (Hamilton, et al., Cancer Research 1983).

Major concerns

Cluster numbers

Leiden based clustering implemented by Seurat is strongly affected by selection of the following four input parameters: number of PCs, k-value, prune value and resolution value. Altering these required input parameters affects the number of clusters identified. Authors should perform Leiden based clustering multiple times using a range of these parameters and demonstrate that the clustering solution of 12, which is extensively analyzed in the paper, is the most robust clustering solution. Without this assurance, it is difficult to justify the findings.

Effect of inhibiting RAC1 on cell differentiation conclusion

In figure 1B authors list GO terms and pathways significantly associated with upregulated genes from cluster 2. The second highest associated GO term, based on p-value, is "negative regulation of cell differentiation". In figure 2B, authors show that cluster 2 is enriched in cells treated with RAC1 inhibitor. This would suggest that RAC1i treatment increases "negative regulation of cell differentiation", and yet authors conclude that "…treatment with RAC1i alone or in combination with EZH2i increased the cell proportions from 7% to approximately 25% of the entire cell population while EZH2i alone had no effect on this cluster (Figure 2B), suggesting that RACi treatment induced differentiation." This conclusion seems to be contradictory to their data. Authors should explain how their data indicates RAC1i induces differentiation and does not negatively regulate cell differentiation, as cluster 2 is enriched in negative regulators of differentiation.

Minor concerns

Cell numbers

Fig 2B shows changes in the percentages of cells in each cluster based on treatment. The statistics indicate that RAC1 treatment statistically increases the proportion of cells in cluster 2 and reduces the proportion in cluster 4. As these numbers are based on relative proportion, it is important to know if there was a large difference in growth of the cell populations after treatment with the inhibitors. One assumes that the authors submitted equal numbers of viable cells for each condition for sequencing, but it should be reported how much the treatment affected viability and growth.

Line 263 describes cluster 2 as "…which is enriched for genes associated with cell death and differentiation (Figure 2C)". This is possibly a typo, as cluster 2 did not have any cell death pathways associated (Fig 1B). I think authors meant to say, "cell development and differentiation".

The exact number of cells depicted in Fig 2A in each panel should be listed somewhere in the paper, so it is known how many cells were analyzed in each condition.

The authors conclusion that RAC1 inhibition reduces numbers of ovarian cancer stem cells by analyzing more stem cell markers and presenting these findings (eg PROM1/CD133, cKIT/CD117, CD44, the other ALDH genes)

Reviewer #2: To draw a valid conclusion, it is necessary to provide more detailed scRNA-seq data and additional results of validation experiments as specified below.

1. For each cluster of cells in scRNA-seq analysis, the most significant markers across all the samples should be tabulated and presented.

2. Genes contributed to the enriched pathways in each cell cluster (Figure 1B-E) should be tabulated and presented.

3. Validation experiments should be conducted to confirm that RAC1i treatment induces expression of inflammatory genes in OVCAR3 cells. It will be interesting to examine whether RACi-induced expression of CXCL1-3 and 8 can be replicated using other OV cell lines.

4. Genes contributed to the altered pathways by EZH2i and RAC1i in Cluster 2 (Figure 4) should be tabulated and presented. Validation experiments should be conducted to confirm that RAC1i treatment induces SIRT7 expression in OVCAR3 cells and examine whether this effect can be extended to other OV cell lines.

6. PLOS authors have the option to publish the peer review history of their article (what does this mean?). If published, this will include your full peer review and any attached files.

Reviewer #1: No

Reviewer #2: No

---

## [Author Response · Author response to Decision Letter 0]

29 Jun 2022

Reviewer #1:

Overview

Nephew, et al., present data on RNA expression in single cells from the Ovcar3 cell line comparing untreated cells to cells treated with inhibitors of EZH2 and RAC1. This work is a follow up to their previous publication that identified DAB2IP as a tumor suppressor that blocks ovarian cancer stem cells, based on cell line models.

Interesting findings reported include

No major changes in gene expression after treating with the EZH2 inhibitor for 48 hours (Fig 2B). This is quite surprising, as one might expect large scale gene expression changes after blocking this histone methyltransferase, which would then cause differential clustering. (eg, Tiffen, et al., Oncotarget 2015)

RAC1 inhibition has remarkable effects causing upregulation of chemokines, SIRT77 and OVGP1, while downregulating ox/phos genes, Pax8 and stem cell markers ADLH1A1, CD24 and SOX2 in OVCAR3 cell line.

The findings are interesting, although their implications for ovarian cancer are tenuous until they are confirmed in more cell lines and in primary ovarian cancer tissue.

Methods summary

Authors treated the Ovcar3 cell line for 48 hrs with DMSO (control), NSC23766 (RAC1 inhibitor), GSK126 (EZH2 inhibitor), or both inhibitors together and then performed single cell RNAseq (scRNAseq) using the 10X Genomics platform. Data was analyzed using CellRanger and Seurat R packages. Data from ~7 to 10k cells from each treatment were combined and analyzed together. Cells were also treated with JC-1 and analyzed by IFC. The Ovcar3 cell line is a hypotriploid, poorly differentiated papillary adenocarcinoma derived from the ascites of a patient who had been treated with cyclophosphamide, adriamycin, and cisplatin 8 months prior to cell collection in 1982 (Hamilton, et al., Cancer Research 1983).

Major concerns

1. Cluster numbers

Leiden based clustering implemented by Seurat is strongly affected by selection of the following four input parameters: number of PCs, k-value, prune value and resolution value. Altering these required input parameters affects the number of clusters identified. Authors should perform Leiden based clustering multiple times using a range of these parameters and demonstrate that the clustering solution of 12, which is extensively analyzed in the paper, is the most robust clustering solution. Without this assurance, it is difficult to justify the findings.

Response: Thank you to the reviewer for this comment. We agree that there are many parameters that affect clustering. We performed a grid search on the clustering parameters (Schneider I. et al. Journal of Translational Genetics and Genomics 2021) and also determined silhouette scores to quantify the performance of our clustering (Rousseeuw PJ. Mathematics 1987). Based on the silhouette plot (Rebuttal Figure 1), for most of our labelled clusters, most cells had positive scores, which indicated better clustering. In general, the means of the silhouette scores for the clusters were also around or above the mean of scores for the clusters using the highest grid search parameters. However, cluster 5: immune/inflammation related did not perform well even though the CXCL marker genes for this cluster were robust (Figure 3). There is not a standard method for analyzing scRNA-seq data and the results we found were statistically significant so even if the clustering was imperfect, the results are still sound and verified in many cases by additional experiments. Furthermore, many of the updated packages and bioinformatics solutions to optimize the parameters for clustering were not available at the time we performed the initial analysis.

Rebuttal Figure 1. Silhouette scores for clustering strategy used in the manuscript.

2. Effect of inhibiting RAC1 on cell differentiation conclusion

In figure 1B authors list GO terms and pathways significantly associated with upregulated genes from cluster 2. The second highest associated GO term, based on p-value, is "negative regulation of cell differentiation". In figure 2B, authors show that cluster 2 is enriched in cells treated with RAC1 inhibitor. This would suggest that RAC1i treatment increases "negative regulation of cell differentiation", and yet authors conclude that "…treatment with RAC1i alone or in combination with EZH2i increased the cell proportions from 7% to approximately 25% of the entire cell population while EZH2i alone had no effect on this cluster (Figure 2B), suggesting that RACi treatment induced differentiation." This conclusion seems to be contradictory to their data. Authors should explain how their data indicates RAC1i induces differentiation and does not negatively regulate cell differentiation, as cluster 2 is enriched in negative regulators of differentiation.

Response: Thank you to the reviewer for pointing out this potentially confusing point. Regarding the GO analysis, more than 50% of genes used for enrichment in “negative regulation of cell differentiation” are also in the GO term “positive regulation of cell differentiation”. Therefore, while the GO analysis suggests that cluster 2 is enriched for regulation of cell differentiation, the directionality is difficult to determine from the GO analysis alone. In Figure 5, we analyzed the expression of genes associated with OCSCs in our scRNA-seq data in the different treatment groups and performed immunofluorescence for OVGP1, which increases in expression in more differentiated OC cells. These results suggest that RAC1 inhibition increased cell differentiation. Because in Figure 2 we do not yet know the directionality of the change in differentiation, in the revised manuscript we have changed the statement mentioned by the reviewer to “... suggesting that RACi treatment altered differentiation”.

Minor concerns

1. Cell numbers

Fig 2B shows changes in the percentages of cells in each cluster based on treatment. The statistics indicate that RAC1 treatment statistically increases the proportion of cells in cluster 2 and reduces the proportion in cluster 4. As these numbers are based on relative proportion, it is important to know if there was a large difference in growth of the cell populations after treatment with the inhibitors. One assumes that the authors submitted equal numbers of viable cells for each condition for sequencing, but it should be reported how much the treatment affected viability and growth.

Response: As the reviewer suggests, we submitted equal numbers of viable cells for each condition for 10X Chromium single cell processing and library preparation and similar numbers of cells were sequenced for each condition. The number of cells in each treatment group was as follows: DMSO 7530, EZH2i 7651, RAC1i 10207, Combo 8232 (these numbers are now included in Supplementary Table S3). In our previous publication (Zong X, et al. Cancer Research 2020), we determined that 50 μM RAC1i had minimal effect on tumorsphere survival/growth when used alone suggesting that the changes in percentages of cells in clusters 2 and 4 are caused by RAC1i treatment and not by decreased cell viability.

2. Line 263 describes cluster 2 as "…which is enriched for genes associated with cell death and differentiation (Figure 2C)". This is possibly a typo, as cluster 2 did not have any cell death pathways associated (Fig 1B). I think authors meant to say, "cell development and differentiation".

Response: Thank you to the reviewer for pointing out this typo. We have changed the text as suggested.

3. The exact number of cells depicted in Fig 2A in each panel should be listed somewhere in the paper, so it is known how many cells were analyzed in each condition.

Response: We have now included the number of cells depicted in Figure 2A in Supplementary Table S3.

4. The authors conclusion that RAC1 inhibition reduces numbers of ovarian cancer stem cells by analyzing more stem cell markers and presenting these findings (eg PROM1/CD133, cKIT/CD117, CD44, the other ALDH genes)

Response: Thank you to the reviewer for this suggestion. We have now included violin plots for CD44, PROM1/CD133, ALDH1A2, ALDH1A3 in Supplementary Figure 1. These genes were expressed at low levels in all conditions and cKIT/CD117 was not detectable so changes in expression of these genes with treatment could not be determined. We also included plots for additional expressed ALDH isoforms. The expression of these genes decreased across most clusters with RAC1i treatment alone or in combination with EZH2i.

Reviewer #2:

To draw a valid conclusion, it is necessary to provide more detailed scRNA-seq data and additional results of validation experiments as specified below.

1. For each cluster of cells in scRNA-seq analysis, the most significant markers across all the samples should be tabulated and presented.

Response: Thank you to the reviewer for this suggestion. We have now included the most significant marker genes for each cluster in Supplementary Table S1. When there were greater than 10 genes enriched in a given cluster only the top 10 were included.

2. Genes contributed to the enriched pathways in each cell cluster (Figure 1B-E) should be tabulated and presented.

Response: As requested, in the revised manuscript we have included Supplementary Table S2, which includes lists of the genes that contributed to the enriched pathways in each cluster. 

3. Validation experiments should be conducted to confirm that RAC1i treatment induces expression of inflammatory genes in OVCAR3 cells. It will be interesting to examine whether RACi-induced expression of CXCL1-3 and 8 can be replicated using other OV cell lines.

Response: In the revised manuscript, we now include expression data for CXCL1-3 in bulk populations of OVCAR3 cells (Supplementary Figure S1A). CXCL1 expression increased with RAC1 inhibition but changes in CXCL2 and CXCL3 expression were not detectable in the bulk population. These findings are consistent with the increase in CXCL1 expression following RAC1i occurring in many clusters whereas CXCL2, CXCL3, and CXCL8 only had increased expression in a small subpopulation of cells (Figure 3). The results also emphasize the importance of performing scRNA-seq, which can identify important changes in gene expression that occur in only a small proportion of the total population of cells. We have not assayed RAC1i-induced changes in CXCL gene expression in other ovarian cancer cell lines as we would be assaying changes in bulk gene expression, which may not reflect changes at the single cell level.

4. Genes contributed to the altered pathways by EZH2i and RAC1i in Cluster 2 (Figure 4) should be tabulated and presented. Validation experiments should be conducted to confirm that RAC1i treatment induces SIRT7 expression in OVCAR3 cells and examine whether this effect can be extended to other OV cell lines.

Response: As requested, we have now included Supplementary Table S4, which lists the genes in cluster 2 that contributed to the pathway enrichment in IPA analysis in Figure 4. We have also confirmed that RAC1i treatment induces SIRT7 expression in bulk OVCAR3 cells (Supplementary Figure S1B). We have not assayed RAC1i-induced changes in SIRT7 gene expression in other ovarian cancer cell lines as we would be assaying changes in bulk gene expression, which may not reflect changes at the single cell level.

---

## [Editor Report · Decision Letter 1]

4 Jul 2022

Single-cell Analysis of a High-grade Serous Ovarian Cancer Cell Line Reveals Transcriptomic Changes and Cell Subpopulations Sensitive to Epigenetic Combination Treatment

PONE-D-22-09003R1

Dear Dr. Nephew,

We’re pleased to inform you that your manuscript has been judged scientifically suitable for publication and will be formally accepted for publication once it meets all outstanding technical requirements.

Kind regards,

Lawrence M. Pfeffer, PhD

Academic Editor

PLOS ONE

Additional Editor Comments (optional):

The authors have been highly responsive to the reviewers and the manuscript is markedly improved. It now merits publication in PLOS One
---

## [Editor Report · Acceptance letter]

25 Jul 2022

PONE-D-22-09003R1 

Single-cell Analysis of a High-grade Serous Ovarian Cancer Cell Line Reveals Transcriptomic Changes and Cell Subpopulations Sensitive to Epigenetic Combination Treatment 

Dear Dr. Nephew:

I'm pleased to inform you that your manuscript has been deemed suitable for publication in PLOS ONE. Congratulations! Your manuscript is now with our production department. 

Kind regards, 

on behalf of

Dr. Lawrence M. Pfeffer 

Academic Editor

PLOS ONE